# The Significance of True Knot of the Umbilical Cord in Long-Term Offspring Neurological Health

**DOI:** 10.3390/jcm10010123

**Published:** 2020-12-31

**Authors:** Yael Lichtman, Tamar Wainstock, Asnat Walfisch, Eyal Sheiner

**Affiliations:** 1Department of Obstetrics and Gynecology, Soroka University Medical Center, Ben-Gurion University of the Negev, Beer-Sheva 8410501, Israel; 2Department of Public Health, Faculty of Health Sciences, Ben-Gurion University of the Negev, Beer-Sheva 8410501, Israel; wainstoc@bgu.ac.il; 3Department of Obstetrics and Gynecology, Hadassah Mt. Scopus Medical Center, Jerusalem 9112001, Israel; asnatwalfisch@yahoo.com

**Keywords:** true knot of cord, perinatal outcomes, long-term neurological morbidity, intrauterine hypoxia

## Abstract

We aimed to study both the short- and long-term neurological implications in offspring born with confirmed knotting of the umbilical cord—“true knot of cord”. In this population based cohort study, a comparison of perinatal outcome and long-term neurological hospitalizations was performed on the basis of presence or absence of true knot of cord. A Kaplan–Meier survival curve was constructed to compare the cumulative incidence of neurological hospitalizations between the study groups. Multivariable regression models were used to assess the independent association between true knot of cord, perinatal mortality and long term neurological related hospitalizations, while controlling for potential confounders. The study included 243,639 newborns, of them 1.1% (*n* = 2606) were diagnosed with true knot of the umbilical cord. Higher rates of intrauterine fetal demise (IUFD) were noted in the exposed group, a finding which remained significant in the multivariable generalized estimation equation, while controlling for confounders. The cumulative incidences of neurological hospitalizations over time were comparable between the groups. The Cox regression confirmed a lack of association between true knot of cord and total long term neurological related hospitalizations. While presence of true knot of the umbilical cord is associated with higher IUFD rates, in our population, however, its presence does not appear to impact the long term neurological health of exposed offspring.

## 1. Introduction

Knotting of the umbilical cord—“true knot of cord”—is a rather rare event (about 1% of term deliveries) [1] and a challenging antepartum diagnosis [2]. Factors predisposing the formation of cord knots include polyhydramnios and multi-parity (due to uterine laxity), as well as diabetes and preterm delivery, all of which enable exaggerated fetal movements [3,4,5]. Other factors that have been associated with true knots are male fetuses and long cords, probably due to the fact that these two often coexist [4,6,7]. The pathophysiology of knotting of the cord is probably a combination of uterine laxity, exaggerated fetal movement and increased amount of amniotic fluid relative to fetal size.

The exact timing of formation of these cord knots is a matter of debate—some argue that knotting of the umbilical cord takes place early in the antenatal course during the late first trimester due to increased amniotic fluid volume/fetal size ratio, while others think that this event mainly takes place during labor [8]. Attempts to diagnose this condition antepartum have been disappointing [2], even with the latest advancements in Doppler sonography, and most cases are recognized only postpartum [9].

The significance of true cord knot is controversial; while several studies show an association with devastating perinatal outcomes (such as intrauterine fetal demise (IUFD), meconium-stained amniotic fluid (MSAF), and low Apgar scores) [4,10,11,12], others have failed to establish any clinical significance [13,14]. Some studies link true cord knots to low birth weight [15], potentially resulting from chronic intrauterine hypoxia [16], while others show no such association [11]. This might be explained by the level of tightness of the knot, also affected by the protection of Wharton’s jelly [17].

Although immediate obstetrical outcomes related to cord knots have been extensively studied, much less is known regarding its long term significance. Our work aimed to shed light on the potential long term neurological impact of true cord knots in exposed offspring.

## 2. Experimental Section

This was a population based retrospective cohort study conducted at the Soroka University Medical Center (SUMC) between the years 1991–2014. SUMC is a sole tertiary medical center located in the Negev region of Israel, which spreads over 60% of Israel’s territory with a population of 730,000 inhabitants in 2017 (and constantly increasing) [18].

Currently, SUMC providing tertiary medical services to about 1,190,000 individuals (composing around 14% of Israel population). The study was approved by the institutional review board (SUMC IRB) and is based on nonselective population data.

The Bedouin Arabs of the Negev are a Muslim society [19]. Bedouin culture places great importance on family and high fertility is central in this society [20]. Thus, multiparity is common [21]. The available prenatal diagnostic services are underused by this population, possibly owing to religious restrictions [22,23], distrust of conventional medical care providers and facilities, geographical distance to healthcare services including available prenatal care services, and patriarchal restriction of female autonomy [19,24].

The primary exposure was presence of true knot of cord, as recorded postpartum by the midwife attending the delivery, in the computerized as well as paper perinatal records which are constantly being revised by professional hospital secretaries for error. The outcome measures included immediate obstetrical outcomes as well as any neurological related hospitalization of the offspring up to 18 years of age, as evident by any neurological diagnosis mentioned in the patients files upon admission to SUMC (for any reason). This was defined as having one diagnosis or more from a pre-defined list of ICD-9 neurological codes detailed in the Appendix A (Table A1). Multiple gestations and congenital malformations cases were excluded from all analyses. If a cord description was missing from the record, it was excluded from the analysis. Follow up time was defined as time to an event or censoring. An event was defined as hospitalization with a neurological diagnosis, including all the non-neurological admissions in which a neurological diagnosis (chronic or acute), was designated for the offspring. Censoring occurred either as death during any hospitalization (other than neurological related), end of study period (January 2014), or when the child reached 18 years of age (calculated according to date of birth). Only the first admission with a neurological related diagnosis for each child was included in the analyses.

Data were collected from two databases that were cross-linked and merged: the computerized hospitalization database of SUMC (“Demog-ICD9”), and the computerized perinatal database of the SUMC obstetrics and gynecology department. The Demog-ICD9 database includes demographic information and ICD-9 codes for all medical diagnoses made during hospitalizations at SUMC. The perinatal database consists of information recorded immediately following delivery by an obstetrician or a midwife. Experienced medical secretaries routinely review the information prior to entering it into the database to insure its maximal completeness and accuracy. Furthermore, the perinatal database was regularly tested and validated by the Department of Epidemiology, Ben-Gurion University of the Negev, Beer Sheva, Israel. Coding is performed after assessing medical prenatal care records as well as routine hospital documents.

Screening for neurological morbidity in the hospital setting is done in the Institute for Child Development which provides diagnostic services, treatment, and follow up, for children with developmental disorders up to age 6 years. The Institute for Child Development has close ties with other ambulatory services and as a consequence, even though the Institute for Child Development’s diagnoses are not part of the SUMC hospitalization database, they are often presented as background diagnoses of the child upon hospitalization. Early assessment of developmental difficulties and disorders occur in Israel routinely at community. If additional evaluation is needed, the children and their families are referred to Child and Family Developmental Centers, where the child is been evaluated. When a child previously diagnosed in a community clinic is being admitted to the hospital, his previous diagnoses are usually exported to the SUMC data base. Additionally, the community clinic and SUMC share the same online interface, which facilitates the process of exporting diagnoses upon admission [25].

Statistical analysis was performed using the SPSS package, 23rd edition (IBM/SPSS, Chicago, IL, USA). Differences in categorical data were assessed by chi-square for general association. T-test was used for comparison of continuous variables with normal distribution. Kaplan–Meier survival analysis was used to compare the cumulative incidence of neurological related hospitalizations over time, up to 18 years of age. A multivariable generalized estimation equation model was used to study the association between true knot of the cord and perinatal mortality. Cox proportional hazards analysis was used to assess a possible independent association between true knot of cord and long term neurological related hospitalizations of the offspring. Both multivariable models adjusted for potential confounding variables and clinically relevant characteristics. These included: gestational age, small for gestational age (SGA, <5th percentile of birthweight according to gestational age and gender), ethnicity, smoking status, maternal diabetes and hypertension. A *p* value of < 0.05 (two sided) was considered statistically significant.

## 3. Results, Figures and Tables

### Results

During the study period, 243,639 newborns met the inclusion criteria. Of them, 1.1% (*n* = 2606) were diagnosed with confirmed true knot of the umbilical cord. Maternal characteristics and pregnancy outcomes in both groups are shown in Table 1. Parturient with true knot of cord were significantly more likely to be multiparous, suffer from hypertension, diabetes, undergo labor induction, and deliver preterm (<37 0/7 weeks’ gestation). Deliveries were more likely to involve meconium stained amniotic fluid (MSAF), and to end with cesarean delivery. Newborns in the exposed group exhibited higher rates of low (<7) Apgar scores, SGA infants, and IUFD.

In the multivariate generalized estimating equation models an independent and significant association was found between presence of a true cord knot and IUFD, while adjusting for ethnicity, smoking status, maternal diabetes, maternal hypertension and offspring date of birth (adjusted odds ratio 3.606; 95% CI 2.685–4.841, *p* < 0.001; Table 2).

For the long-term neurological morbidity analyses, perinatal mortality cases were excluded, leaving 242,342 newborns, 1.1% (2558) of which were exposed. During the 22 year follow up period (up to the age of 18), total neurological hospitalization rates were comparable between the groups (3.7% in the exposed group and 3.1% in the comparison group, *p* = 0.078; Table 3) as were the cumulative incidences of neurological hospitalizations over time (log rank *p* = 0.12; Figure 1). Attention deficit disorders associated with hospitalizations were slightly more common in the exposed group (0.16% vs. 0.06% in controls, *p* = 0.041).

The Cox regression model confirmed a lack of association between true knot of cord and total long term neurological related hospitalizations (adjusted HR = 1.236, 95% CI 0.728–2.1, *p* = 0.432; Table 4), as well as specifically for attention deficit disorders (adjusted HR 2.6, 95% CI 0.96–7.04, *p* = 0.06). The Cox model adjusted for diabetes, hypertensive disorders, maternal age and offspring date of birth. In a sensitivity analysis, the groups were stratified according to gestational age at delivery into term deliveries (37.0 weeks or more) and preterm deliveries (less than 37.0 weeks). The results remained similar (adjusted HR = 1.13, 95% CI 0.91–1.41, *p* = 0.261 for term deliveries and adjusted HR = 1.27, 95% CI 0.75–2.16, *p* = 0.365 for preterm deliveries). 

## 4. Discussion

In this large retrospective cohort study with a long follow up period, we found increased rates of adverse obstetrical outcomes in pregnancies associated with true knot of cord, and specifically with intra-uterine fetal demise, as well as low Apgar scores, preterm deliveries, cesarean deliveries, and meconium stained amniotic fluid. However, in the long term perspective, no association was found between true knot of cord and long term adverse neurological outcome (involving hospitalization) in the offspring, up to 18 years of age.

The increased rates of pretem delivery (PTD), cesarean delivery (CD), and low Apgar scores can potentially be explained by the association of true cord knots with non-reassuring fetal heart rate (NRFHR) and MSAF, thus predisposing these deliveries to iatrogenic interventions resulting in preterm deliveries, cesarean delivery [4,26] and low Apgar scores [27]. 

In addition, MSAF, polyhydramnios, true knot of cord and hypertensive disorders of pregnancy were all found to associated with IUFD [12], which can explain the significantly increased rate of IUFD in the exposed group. The association of true cord knots with IUFD appears to be significant and independent in the regression model, which was meticulously controlled for multiple confounders. In light of the severity of the immediate adverse outcomes reinforced by our study, it appears that increased antenatal surveillance is appropriate, in cases where a true knot of cord is diagnosed antenatally. It may also be appropriate to screen for it in high risk populations, if a reliable screening method was available. 

In contrast to the clear adverse impact of true knot exposure on perinatal outcome, our data conformed a lack of association between true cord knots and long-term neurological morbidity (associated with hospitalizations) in the offspring. To the best of our knowledge, no studies have previously focused on the long-term impact of true cord knots. We hypothesized that fetuses exposed to true knot of cord may have suffered some degree of hypoxemia during the pregnancy or labor process thus predisposing them to long term adverse neurological consequences. However, the results of this work suggest otherwise. True knot of cord may act in a severity dependent manner, meaning that the damage caused by the presence of the cord knot depends on the degree of venous flow obstruction caused by it, in a way that a tight knot may cause acute hypoxia, leading to immediate adverse outcome like IUFD; while a looser knot may result in chronic mild hypoxia and a less devastating outcome. In this manner, some or even most fetuses with knots might not be effected at all.

Several weaknesses of the study must be acknowledged:Although several confounders were controlled for and an independent association was found with IUFD, it is possible due to the retrospective nature of the study, that some confounders were not accounted for.Most childhood neurological morbidities, especially on the “lighter” side of the spectrum, are cared for in an ambulatory setting and were not accounted for in this long-term analysis. This can lead to under reporting of some diagnosis due to the fact that some diagnosed children are not hospitalized. Furthermore, for several of the outcomes (like autistic spectrum disorders), diagnosis typically only comes through specialized screening, which is a potential for selection bias (of children who suffer from the condition but were not screened for it). Nevertheless, some of the conditions included in the study are significant morbidities, and therefore are likely to necessitate hospitalization at some point. There is a possibility that the study groups were underpowered to detect neurological-related hospitalizations in the offspring.Hospitalization at a different, distant, medical center, although unlikely, is possible. SUMC is the only tertiary center in the Negev region, it is reasonable to assume that this is the only place for children to be hospitalized in case of morbidity; however, there can be no guarantee of that. Therefore, ascertainment bias potentially exists. There seems to be no reason, however, for either of those phenomenon to be more common in either of the compared groups.It was assumed that children that did not visit our hospital were healthy (which might be a biased assumption). This possibility as well is probably just as likely in both the exposed and unexposed groups.A heterogeneous group of neurological outcomes was used rather than a specific neurological diagnosis. The purpose of this work was to search for an association between different groups of neurological morbidities and true knot of cord upon birth. We did not look for specific diagnoses since no specific associations were mentioned in the literature nor were part of our hypothesis. Additionally, these types of diagnoses are quite rare and looking for specific diagnoses (rather than groups of diagnoses) would have diminished the power of our results.

To conclude, the results of this large population based study with a long follow up period contribute some knowledge to the understating of the significance of true knot of cord. Although associated with elevated rates of IUFD, in our population, however, no severe long term neurological impact was noted.

## Figures and Tables

**Figure 1 jcm-10-00123-f001:**
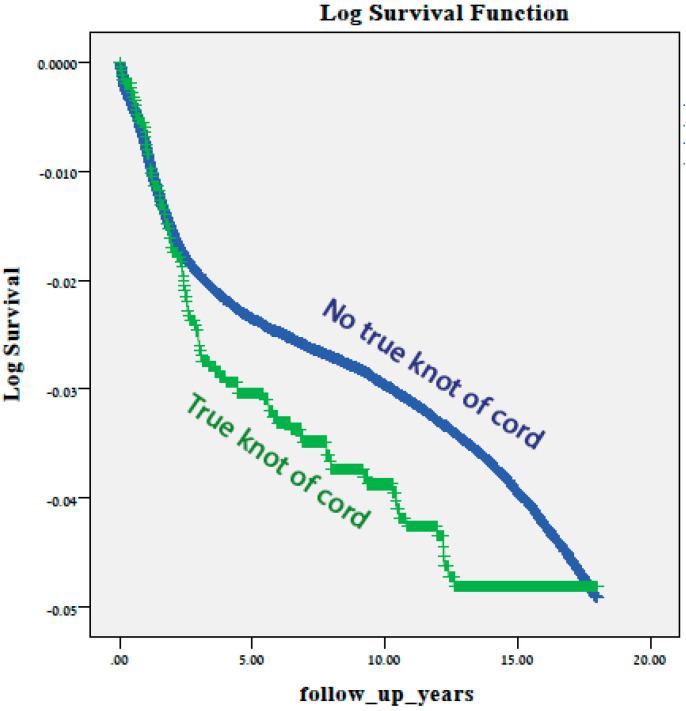
Log of survival, total neurological hospitalizations up to the age of 18 years by presence or absence of true knot of cord * (* log rank test *p* = 0.120).

**Table 1 jcm-10-00123-t001:** Perinatal outcome according to presence or absence of true knot of the umbilical cord.

Perinatal Outcomes		True Knot of Cord % (*n* = 2606) *	No True Knot of Cord % (*n* = 241,076)	Odds Ratio (Confidence Interval)	*p-*Value	
Ethnicity	Jewish	60 (1572)	47.2 (113,782)			<0.001
	Bedouin	39.7 (1034)	52.8 (127,294)			<0.001
Mean maternal age (years, mean ± SD)		30.2 ± 5.9	28.1 ± 5.8			<0.001
Parity	1	15.5 (405)	23.7 (57,100)			<0.001
	2–4	54.8 (1428)	51.1 (123,086)			
	≥5	29.7 (773)	25.2 (60,837)			
Maternal Diabetes		8.4 (220)	5 (11,939)		1.77 (1.539–2.034)	<0.001
Maternal Hypertension		7.4 (192)	5 (12,055)		1.511 (1.303–1.752)	<0.001
Mean gestational age (weeks, mean ± SD **)		38.8 ± 2.4	39.1 ± 1.09			<0.001
Preterm delivery (<37 0/7 weeks of gestation)		10.5 (274)	6.8 (16,446)		1.605 (1.415–1.821)	<0.001
Induced labor		28.7 (747)	26.1 (62,897)		1.138 (1.045–1.24)	0.003
Cesarean delivery		17.4 (453)	13.5 (32,573)		1.347 (1.216–1.491)	<0.001
Placental abruption		0.8 (21)	0.6 (1338)		1.456 (0.944–2.244)	0.087
Meconium stained amniotic fluid		18.9 (493)	14.7 (35,399)		1.356 (1.228–1.496)	<0.001
Low (<7) 1 min Apgar score		7.3 (190)	5.3 (12,800)		1.403 (1.209–1.627)	<0.001
Low (<7) 5 min Apgar score		2.9 (76)	2.3 (5433)		1.303 (1.035–1.639)	0.024
Perinatal mortality	Total perinatal mortality	1.8 (48)	0.5 (1292)		3.483 (2.604–4.658)	<0.001
	Intra uterine	1.5 (39)	0.3 (713)		5.122 (3.702–7.086)	<0.001
	Intra-partum	0.1 (2)	0.024 (60)		3.085 (0.754–12.629)	0.099
	Immediately post-partum	0.3 (7)	0.2 (519)		1.248 (0.592–2.634)	0.560
Mean birth weight (grams, mean ± SD)		3209 ± 564	3205 ± 510			0.73
Low birth weight (<2500 g)		9.2 (239)	6.7 (16,165)		1.405 (1.229–1.606)	<0.001
Male gender		61.6 (1604)	50.7 (122,273)		1.555 (1.437–1.684)	<0.001
Female gender		38.4 (1002)	49.3 (118,803)		1.555 (1.437–1.684)	<0.001

* All numbers presented in % (*n*) unless otherwise stated, ** SD = Standard deviation.

**Table 2 jcm-10-00123-t002:** Multivariable regression analysis for the association between true knot of cord and perinatal mortality.

	Adjusted Odds Ratio (Confidence Interval)	*p-*Value
**True knot of cord**	3.606 (2.685–4.841)	<0.001
**Ethnicity (Jewish compared to Bedouin)**	0.595 (0.529–0.668)	<0.001
**Smoking**	1.52 (0.909–2.54)	0.11
**Maternal diabetes**	0.628 (0.463–0.852)	0.003
**Maternal Hypertension**	2.089 (1.733–2.518)	<0.001
**Birth year**	0.937 (0.929–0.946)	<0.001

**Table 3 jcm-10-00123-t003:** Long term neurological hospitalizations of the offspring born with and without true knot of the umbilical cord.

Neurological Morbidity	True Knot of Cord % (*n* = 2558)	No Knot of Cord % (*n* = 239,784)	*p-*Value
**Autistic spectrum disorders**	0.0003 (1)	0.0001 (27)	0.193
**Eating disorders**	0.2 (6)	0.2 (429)	0.508
**Sleeping disorders**	0.0003 (1)	0.0001 (47)	0.486
**Movement disorders**	2.2 (56)	1.8 (4416)	0.194
**Cerebral palsy**	0.1 (2)	0.1 (199)	0.933
**Psychiatric emotional**	0.5 (12)	0.5 (1183)	0.862
**Attention deficit disorders**	0.2 (4)	0.1 (139)	0.041
**Developmental disorders**	0.2 (5)	0.1 (234)	0.117
**Degenerative, demyelination**	0.03 (1)	0.1 (180)	0.508
**Headache**	0 (0)	0.0002 (54)	0.448
**Myopathy**	0.1 (2)	0.1 (136)	0.651
**Other**	0.4 (10)	0.4 (907)	0.917
**Total Neurological hospitalizations**	3.7 (95)	3.1 (7448)	0.078

**Table 4 jcm-10-00123-t004:** Cox regression analysis for the association between long term neurological morbidity and true knot of cord.

	Adjusted Hazard Ratio (Confidence Interval)	*p* Value
**True knot of cord**	1.236 (0.728–2.1)	0.432
**Diabetes**	1.143 (0.87–1.501)	0.337
**Hypertension**	1.249 (1.017–1.534)	0.034
**Maternal age (at birth)**	0.993 (0.981–1.005)	0.265
**Child birth year**	1.092 (1.076–1.109)	<0.001

## Data Availability

According to the local Helsinky guidelines data cannot be provided outside of hospital.

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
