# Peer review of "The Significance of True Knot of the Umbilical Cord in Long-Term Offspring Neurological Health"

_jcm, 2020, doi:10.3390/jcm10010123_

Round 1
Reviewer 1 Report
The authors present a large population based cohort, aimed to study the short and long term neurological implications in offspring born with confirmed true knot of umbilical cord. They found that while true knot of the umbilical cord was associated with higher IUFD rates, it did not appear to impact the long term neurological health of exposed offspring.
This study is very interesting and well written. The topic is a matter of controversy and the long-term implications were never investigated. It is based on a large population cohort with follow up.
I believe that this would be interesting to the readers and certainly recommend publication.
I have minor comments to the authors.
- Apgar is a name, so please change APGAR to Apgar.
- Authors compared two ethnicities, and controlled for it; please expand on Bedouin ethnicity (I would describe the population in the Methods section). Not all readers are familiar with it.
- The study duration is long (between the years 1991-2014)- practice have changes during this period. Accordingly, I would add the year of birth to the Cox model.
Author Response
Reviewer 1:
Comments:
- Apgar is a name, please change APGAR to Apgar.
- The correction was made in the text as is appropriate.
- Authors compared two ethnicities, and controlled for it; please expand on Bedouin ethnicity (I would describe the population in methods section). Not all readers are familiar with it.
- A descriptive paragraph about the Bedouin community was added to the materials and methods section:
- The Bedouin Arabs of the Negev are a Muslim society. The Bedouin culture places great importance on family, and high fertility is central in this society (Pregnancy Termination among Bedouin Couples Referred to Third Level Ultrasound Clinic. J. Obstet. Gynecol. Reprod. Biol. 1998, 76 (2), 141–146).
- Thus, multiparity is common (Sheiner, E.; Shoham-Vardi, I.; Ohana, E.; Segal, D.; Mazor, M.; Katz, M. Characteristics of Parturients Who Choose to Deliver without Analgesia. Psychosom. Obstet. Gynaecol. 1999, 20 (3), 165–169).
- The available prenatal diagnostic services are underused by this population, possibly owing to religious restrictions, distrust of conventional medical care providers and facilities, geographical distance to healthcare services including available prenatal care services, and patriarchal restriction of female autonomy (Estis-Deaton, A.; Sheiner, E.; Wainstock, T.; Landau, D.; Walfisch, A. The Association between Inadequate Prenatal Care and Future Healthcare Use among Offspring in the Bedouin Population. J. Gynecol. Obstet. 2017, 139 (3), 284–289).
- The study duration is long (between the years 1991-2014) – practice have changed during this period. Accordingly, I would add the year of birth to the Cox model.
- Thank you for the comment. According to this insightful comment we have added the year of birth of the offspring to our COX model.
|
|
Adjusted hazard ratio (Confidence Interval) |
p value |
|
True knot of cord |
1.236 (0.728-2.1) |
0.432 |
|
Diabetes |
1.143 (0.87-1.501) |
0.337 |
|
Hypertension |
1.249 (1.017-1.534) |
0.034 |
|
Maternal age (at birth) |
0.993 (0.981-1.005) |
0.265 |
|
Birth year |
1.092 (1.076-1.109) |
<0.001 |

Reviewer 2 Report
this is an intersting manuscript dealing with true knots impact on short and long term neonatal outcome.
The results concerning impact on perinatal morbitidy is important especially because of the large size of the sample but is not new nor original.
The single center study limits the conclusion.
Some important questions regarding the methodology:
How to be sure of the normal neurologic developpement of the children. The children can be refeered to an other center or can have non diagnosed disabilities.
How many children had a neurologic examination? in the cohorte?
What classification of neurological disabilities has been used?
How many children were lost to follow up? How is organized the link between the mother medical chart and the children medical chart? Is this link Always efficient or available?
How many cords descriptions were missing ?
How many multiple pregnancies? chorionicity can impacts on the short and long terme outcomes this information need to be add in the manuscript or exclusion of twins pregnancies of multivariate analysis on this point.
table 3 the overall number of babies is 242 342 .....it's mean that there is no lost to follow up, this is impossible. Moreover how long were the children follow up? it's also mean that the children with no classification of neurologic disabilities were considered as normal and this is not scientifically suitable.
The limits also of the codification should be further discussed
Author Response
Reviewer 2:
- How to be sure of the normal neurologic developpement of the children. The children can be refeered to an other center or can have non diagnosed disabilities.
- As mentioned, because SUMC is the only tertiary medical center in the Negev, which comprises around 60% of Israel’s territory, it is very unlikely that a significant percentage of the children were visiting a different medical center in the follow up period (unless they were referred for a specific reason). This is why loss to follow up is probably minimal.
- How many children had a neurologic examination? in the cohorte?
- We thank the reviewer for the comment. We do not have data on specific neurological examination. Nevertheless, since all admitted patients are examined by a doctor at least once on admission, it is very likely that all of the participants had gone a neurological examination during their physical examination on admission.
- What classification of neurological disabilities has been used?
- ICD-9-CM (International Classification of Diseases, Ninth Revision) coding system was used which is the official system of assigning codes to diagnoses and procedures associated with hospital utilization in the United States and Israel among other countries. Neurological outcomes assessed included offspring hospitalisations up to the age of 18 years due to primary or secondary (ie not necessarily the primary diagnosis for the current hospitalisation) neurological morbidities. These included as having at least one diagnosisof the following: movement disorders, cerebral palsy, autistic spectrum disorders, eating disorders, psychiatric disease, attention deficit hyperactivity disorder, and developmental disorders. The predefined ICD‐9 code list of all diagnoses included in each of these conditions is detailed in the Supplemental Table to the manuscript.
- How many children were lost to follow up? How is organized the link between the mother medical chart and the children medical chart? Is this link Always efficient or available?
- As mentioned above, loss to follow up is minimal due to the character of health care facilities in the Negev. Data for the present analysis were derived from two computerized databases that had been cross-linked and merged. The SUMC perinatal database included information recorded by an obstetrician immediately after delivery. Subsequently, medical secretaries routinely review the information for completeness and accuracy before it is entered into the database. After evaluating prenatal care records together with the routine hospital documents, coding is performed. These measures ensure maximal completeness and accurateness of the databases. Furthermore, the perinatal database was regularly tested and validated by the Department of Epidemiology, Ben-Gurion University of the Negev, Beer Sheva, Israel. The general SUMC hospitalization database includes demographic information and International Classification of Diseases, ninth revision codes (ICD‐9), for all medical diagnosis made during any hospitalization. All newborns are issued with a national security ID, which is then registered in the mother's formal identification card. These identification numbers can never be changed nor duplicated within the population. This allowed us to be certain of the association between all mothers and children in our datasets.
- How many cords descriptions were missing?
- When the attending physician or midwife found true knot of cord after delivery, it was coded as "case". If no true knot of cord was recorded it was considered as part of our comparison group.
- How many multiple pregnancies? chorionicity can impacts on the short and long terme outcomes this information need to be add in the manuscript or exclusion of twins pregnancies of multivariate analysis on this point.
- We excluded multifetal pregnancies (n = 11 454) and newborns with congenital and structural anomalies (n = 5323). This information was added to the article.
- table 3 the overall number of babies is 242 342 .....it's mean that there is no lost to follow up, this is impossible. Moreover how long were the children follow up? it's also mean that the children with no classification of neurologic disabilities were considered as normal and this is not scientifically suitable.
- Thank you for your comment. Indeed the assumption we made was that there is no lost to follow up. This is probably true or very close to the truth since SUMC is the only tertiary center in the Negev region, it is reasonable to assume that this is the only place for children to be hospitalized in in case of morbidity, however, there can be no guarantee of that. Therefore, ascertainment bias potentially exists. It was added to the limitations of the article.
- There seems to be no reason, however, for either of those phenomena to be more common in either of the compared groups. If the ascertainment bias is, in fact, non‐differential, it is a conservative bias that may underestimate the true risk associated with offspring of mothers with true knot of cord.
- The limits also of the codification should be further discussed.
- We thank the reviewer for the comment and accordingly we have added the following paragraphs to the discussion section of the article: "Screening for neurological morbidity in the hospital setting is done in the Institute for Child Development which provides diagnostic services, treatment, and follow up, for children with developmental disorders up to age 6 years. The Institute for Child Development has close ties with other ambulatory services and as a consequence, even though the Institute for Child Development's diagnoses are not part of the SUMC hospitalization database, they are often presented as background diagnoses of the child upon hospitalization. Early assessment of developmental difficulties and disorders occur in Israel routinely at community. If additional evaluation is needed, the children and their families are referred to Child and Family Developmental Centers, where the child is been evaluated. When a child previously diagnosed in a community clinic is being admitted to the hospital, his previous diagnoses are usually exported to the SUMC data base. Additionally, the community clinic and SUMC share the same online interface, which facilitates the process of exporting diagnoses upon admission (Pariente, G.; Wainstock, T.; Walfisch, A.; Landau, D.; Sheiner, E. Placental Abruption and Long-Term Neurological Hospitalisations in the Offspring. Perinat. Epidemiol. 2019, 33 (3), 215–222).
- The cohort was designed using two databases that were crosslinked and combined (based on mothers and infants identification numbers); the perinatal and the hospitalization databases of SUMC.
- Several of the neurological conditions evaluated in this study are usually diagnosed and treated in an ambulatory setting. This can lead to under reporting of some diagnosis due to the fact that some diagnosed children are not hospitalized . Furthermore, for several of the outcomes (like autistic spectrum disorders), diagnosis typically only comes through specialized screening, which is a potential for selection bias (of children who suffer from the condition but were not screened for it). Nevertheless, some of the conditions included in the study are significant morbidities, and therefore are likely to necessitate hospitalization at some point. (Pariente, G.; Wainstock, T.; Walfisch, A.; Landau, D.; Sheiner, E. Placental Abruption and Long-Term Neurological Hospitalisations in the Offspring. Perinat. Epidemiol. 2019, 33 (3), 215–222).

Reviewer 3 Report
Overall, well written study. A topic worth reviewing given its multiple considerations and increasing relevance to modern practice. It would be pertinent to get the point to the reader. There are a few minor revisions necessary.
Author Response
Thank you for your comment
No revisions were required by you.
Reviewer 4 Report
This is a study on an interesting subject, namely the long term consequences of a knot on the umbilical cord. Unfortunately, as specified in my comments to the authors, I have several major concerns.
Abstract
- The abstract is not very informative. The total number of births is stated, but not the number of fetuses with umbilical cord knots. No effect measures are mentioned.
- It is questionable if the conclusion that cord knots does not impact the long term neurological health of exposed offspring is supported by the presented data.
Introduction
- The introduction is short, but I have no comments.
Experimental section (seem to be what in most journals is labelled “Maternal and Methods”
- It is not clearly stated how “true knot” was defined, and how it was reported in the patient journals. No Icd-code was supplied.
- Censoring occurred either as death during any hospitalization – does this mean that deaths that did not occur in the actual hospital were not recognized? If so – this is a major draw-back of the study.
- The main outcome, “any neurological diagnosis” was defined as having one diagnosis or more from a very long list of f ICD-9 neurological codes. With low numbers, this kind of extremely heterogenious composit outcomes is sometimes used, but it should be recognized as a major shortcoming of the study.
Results
- The data shown in Table 1 makes me wonder if the births taking place in the specific hospital is representative for other births. It seems strange that about 30% and 25% of women with cord knots, or without, respectively, were of parity 5 or more. Or is this distribution common in Israel?
- It is said that table two shows a multivariate analysis. I guess that the authors mean a multivariable one?
- Table 2 again…. What is the rationale for adjusting for SGA when investigating perinatal death? Surely, SGA must be considered as an intermediate outcome – on the pathway between umbilical cord complications and perinatal outcome?
- The reported rates of neurological morbidity are suspiciously low. E.g., only 0.2% and 0.1% in the cord knot group and the controls, respectively. The association was formally statistically significant, but might be heavily biased from underreporting.
- In the Cox-regression analyses, another possible intermediate variable was adjusted for (gestational age).
Discussion
- Given the low prevalence of the various neurological outcomes, I doubt that any conclusions can be made regarding the long term neurological outcome after an umbilical cord knot.
Author Response
Reviewer 4:
This is a study on an interesting subject, namely the long term consequences of a knot on the umbilical cord. Unfortunately, as specified in my comments to the authors, I have several major concerns.
Abstract
- The abstract is not very informative. The total number of births is stated, but not the number of fetuses with umbilical cord knots. No effect measures are mentioned.
- The number of cases with knots was added to the abstract, as was suggested.
- It is questionable if the conclusion that cord knots does not impact the long term neurological health of exposed offspring is supported by the presented data.
According to the comment of the reviewer we have revised the conclusions to: In our population, true knots of cord does not impact the long term neurological health of exposed offspring.
Introduction
The introduction is short, but I have no comments.
Experimental section (seem to be what in most journals is labelled “Maternal and Methods”)
- It is not clearly stated how “true knot” was defined, and how it was reported in the patient journals. No Icd-code was supplied.
- True knot of cord was defined after birth by the midwife or the attending physicians. ICD-9 code for true knot of cord is 90048.
- Censoring occurred either as death during any hospitalization – does this mean that deaths that did not occur in the actual hospital were not recognized? If so – this is a major draw-back of the study.
- Censoring occurred in case of death (during hospitalization, other than neurologically related), at age 18 years, or at the end of the study period. If child's death occurred in hospital, during hospitalization with neurological‐related diagnosis, follow‐up was terminated at hospitalization. Deaths of children that happen outside the hospital are routinely brought to the SUMC, since is the only tertiary hospital in the entire Negev region.
- The main outcome, “any neurological diagnosis” was defined as having one diagnosis or more from a very long list of f ICD-9 neurological codes. With low numbers, this kind of extremely heterogeneous composite outcomes is sometimes used, but it should be recognized as a major shortcoming of the study.
- Thank you for this precise comment. Indeed as you mentioned, a heterogeneous group of outcomes was used rather than a specific diagnosis. The purpose of this work was to search for an association, if exists between different types of neurological morbidities and true knot of cord upon birth. We did not look for specific diagnoses since no such specific associations were mentioned in the literature nor were part of our hypothesis. Additionally, these types of diagnoses are quite rare and looking for specific diagnoses (rather than groups of diagnoses) would have diminished the power of our results.
Results
- The data shown in Table 1 makes me wonder if the births taking place in the specific hospital is representative for other births. It seems strange that about 30% and 25% of women with cord knots, or without, respectively, were of parity 5 or more. Or is this distribution common in Israel?
- Grand -multiparity is quite common in Israel in general and in the Negev population even more so since about 50% of deliveries in SUMC are of women from Bedouin ethnicity. According to the comment of the reviewer we added information to the Methods section about our population and specifically regarding the Bedouin ethnicity, which is characterized by large parity.
- It is said that table two shows a multivariate analysis. I guess that the authors mean a multivariable one?
- Indeed this is a multivariable analysis, and it was corrected as is appropriate.
- Table 2 again…. What is the rationale for adjusting for SGA when investigating perinatal death? Surely, SGA must be considered as an intermediate outcome – on the pathway between umbilical cord complications and perinatal outcome?
- Thank you for the comment. The rationale in adjusting for SGA is in order to demonstrate that the perinatal mortality can be attributed specifically to the presence of true knot of cord and not to the fact that the infants were small for gestational age.
- As mentioned in your further note, another intermediate variable was adjusted for (gestational age).
- Nevertheless, after considering your comment, we have used an additional multivariable regression model to look for an association between true knot of cord and perinatal mortality and a Cox proportional hazard analysis for possible association with long term neurological morbidity. This time without adjusting for the above mentioned confounding factors (namely SGA and preterm delivery). The results did not differ from the original analysis:
|
|
Adjusted Odds Ratio (Confidence Interval) |
p value |
|
True knot of cord |
3.606 (2.685-4.841) |
<0.001 |
|
Ethnicity (Jewish compared to Bedouin) |
0.595 (0.529-0.668) |
<0.001 |
|
Smoking |
1.52 (0.909-2.54) |
0.11 |
|
Maternal diabetes |
0.628 (0.463-0.852) |
0.003 |
|
Maternal Hypertension |
2.089 (1.733-2.518) |
<0.001 |
|
Birth year |
0.937 (0.929-0.946) |
<0.001 |
- The reported rates of neurological morbidity are suspiciously low. E.g., only 0.2% and 0.1% in the cord knot group and the controls, respectively. The association was formally statistically significant, but might be heavily biased from underreporting.
- Thank you for your comment. In our cohort, 1.1% (n=2,606) were diagnosed with confirmed true knot of the umbilical cord. The reported incidence of true knot of cord in the literature, is around 1% ( Spellacy WN, Gravem H, Fisch RO. The umbilical cord complications of true knots, nuchal coils, and cords around the body. Report from the collaborative study of cerebral palsy. Am J Obstet Gynecol. 1966 Apr 15), consequently it seems that no underreporting took place.
- We thank the reviewer for the comment, and accordingly the following paragraph was added to the article: Most childhood neurological morbidities, especially on the “lighter” side of the spectrum, are cared for in an ambulatory setting and were not accounted for in this long-term analysis. Most of the neurological conditions evaluated in this study are usually diagnosed and treated in an ambulatory setting. This can lead to under reporting of some diagnosis due to the fact that some diagnosed children are not hospitalized . Furthermore, for several of the outcomes (like autistic spectrum disorders), diagnosis typically only comes through specialized screening, which is a potential for selection bias (of children who suffer from the condition but were not screened for it). Nevertheless, some of the conditions included in the study are significant morbidities, and therefore are likely to necessitate hospitalization at some point. Since the cohort is rather small; There is a possibility that the study groups were underpowered to detect neurological‐ related hospitalizations in the offspring.
- In the Cox-regression analyses, another possible intermediate variable was adjusted for (gestational age).
- As mentioned above, regarding SGA, the purpose of the adjustment for gestational age was to determine whether the mortality can be attributed to true knot of cord and not to the fact that some infants were born prematurely.
- As mentioned in note number 8, we have constructed a new analysis without adjusting for neither SGA nor Gestational age. The results did not differ ( the model is presented in point number 8).
Discussion
Given the low prevalence of the various neurological outcomes, I doubt that any conclusions can be made regarding the long term neurological outcome after an umbilical cord knot.
According to the comment of the reviewer we have added the comment to the limitation section of the article:
Several weaknesses of the study must be acknowledged: Firstly
- Although several confounders were controlled for and an independent association was found with IUFD, it is possible due to the retrospective nature of the study, that some confounders were not accounted for.
- Second and importantly, most childhood neurological morbidities, especially on the “lighter” side of the spectrum, are cared for in an ambulatory setting and were not accounted for in this long-term analysis. Most of the neurological conditions evaluated in this study are usually diagnosed and treated in an ambulatory setting. This can lead to under reporting of some diagnosis due to the fact that some diagnosed children are not hospitalized . Furthermore, for several of the outcomes (like autistic spectrum disorders), diagnosis typically only comes through specialized screening, which is a potential for selection bias (of children who suffer from the condition but were not screened for it). Nevertheless, some of the conditions included in the study are significant morbidities, and therefore are likely to necessitate hospitalization at some point. Since the cohort is rather small; There is a possibility that the study groups were underpowered to detect neurological‐ related hospitalizations in the offspring.
- Third, hospitalization at a different, distant, medical center, although unlikely, is possible. SUMC is the only tertiary center in the Negev region, it is reasonable to assume that this is the only place for children to be hospitalized in in case of morbidity; however, there can be no guarantee of that. Therefore, ascertainment bias potentially exists. There seems to be no reason, however, for either of those phenomenon to be more common in either of the compared groups. If the ascertainment bias is, in fact, non‐differential, it is a conservative bias that may underestimate the true risk associated with offspring of mothers with true knot of cord.
- It was assumed that children that did not visit our hospital were healthy (which might be a biased assumption). This possibility as well is probably just as likely in both the exposed and unexposed groups.
- A heterogeneous group of neurological outcomes was used rather than a specific neurological diagnosis. The purpose of this work was to search screen for an association between different groups or types of neurological morbidities and in the population of children who have survived an event of true knot of cord upon birth. We did not look for specific diagnoses since no such specific associations were mentioned in the literature nor were part of our hypothesis. Additionally, these types of diagnoses are quite rare and looking for specific diagnoses (rather than groups of diagnoses) would have diminished the power of our results.
To conclude, the results of this large population based study with a long follow up period contribute some knowledge to the understating of the significance of true knot of cord. Although associated with elevated rates of IUFD, in our population, however, no severe long term neurological impact was noted.

Round 2
Reviewer 2 Report
the paper is now suitable for publication